# A Review of High-Fidelity Computational Fluid Dynamics for Floating Offshore Wind Turbines

Shun Xu , Yingjie Xue, Weiwen Zhao and Decheng Wan *

Computational Marine Hydrodynamics Lab (CMHL), School of Naval Architecture, Ocean and Civil Engineering, Shanghai Jiao Tong University, Shanghai 200240, China
* Correspondence: dcwan@sjtu.edu.cn

**Abstract:** The design and development of floating offshore wind turbines (FOWTs) is an attractive issue in the wind energy harvesting field. In this study, the research related to the high-fidelity computational fluid dynamic simulations of FOWTs is comprehensively summarized and analyzed. Specifically, the component-level studies including aerodynamics, aeroelasticity and hydrodynamics are presented. The system studies with increasing complexity are performed, such as the simplified aerodynamics, prescribed platform motions and fully coupled aero-hydrodynamics, as well as a little knowledge relevant to the aero-hydro-elastic behaviors. This study emphasizes that some efforts should shift to the research on strongly coupled aero-hydro-elastic performance of FOWTs with the increasing rotor diameter. Moreover, further investigations of more realistic atmospheric inflows and strong interactions between multi-FOWTs are required. This study aims to introduce the hotspots of high-fidelity simulations of FOWTs to novel researchers, as well as to provide some suggested solutions.

**Keywords:** floating offshore wind turbines; computational fluid dynamics; fully coupled performance; actuator models; blade deformations

## 1. Introduction

In recent years, the development and utilization of offshore wind energy has become a burning issue and many dedicated efforts have focused on it, which has led to great advancements in offshore wind turbines in terms of scheme of design, build, operation and maintenance. Compared to the onshore wind energy harvesting, the benefits of offshore wind energy resources are obvious. For instance, abundant wind resources with higher wind speed and lower turbulence intensity, less constraints for space resources, and no visual and noise pollution, etc. [1]. The Global Wind Energy Council (GWEC) exhibits an outlook of new wind power installations for next five years (2022–2026), it is expected that the new installed offshore wind power in 2026 is 31.4 GW [2]. An increase of 48.8% compared to 21.1 GW in 2021 for offshore wind power is estimated, which is significantly higher than onshore wind power growth of 34.3%. If the good growth for offshore wind power continues, the global offshore wind power will reach up to 228 GW by 2030 and 1000 GW by 2050 [3].

The offshore wind turbines can be divided into two categories, bottom-fixed offshore wind turbines and floating offshore wind turbines (FOWTs). As their names suggest, the foundations of bottom-fixed offshore wind turbines are fixed in the seabed, while the support structures of offshore floating wind turbines are floating and connected to the seabed through mooring lines. One of the primary constraints for bottom-fixed offshore wind turbines is the applicability for water depth < 60 m [4]. When the water depth > 60 m (the deep water), the costs of construction and installation of bottom-fixed foundations increase intensely, which is not commercially applicable. However, more than 80% of offshore wind resources are available in sea area with depth > 60 m. In order to address

the cost issue of bottom-fixed wind turbines and pursue the capture of abundant wind resources in the deep-sea area, the FOWTs have been designed and developed.

The FOWTs can be divided into three major categories based on the types of supporting platforms, that are, spar buoy, semi-submersible and tension leg concepts [5,6], and sometimes also with a barge concept [7]. It is noteworthy that the mooring system can cause considerable costs due to the fact that the length of mooring lines is approximately 4 times the water depth. Moreover, the technology of FOWTs is comparatively immature than that of bottom-fixed offshore wind turbines. Consequently, the levelized cost of energy (LCOE) of FOWTs is unaffordable for full commercial utilization [8]. It is expected that the LCOE of bottom-fixed offshore wind turbines will decrease by 70% from 2015 to 2025, and the similar convergence trend will occur for FOWTs in 2030 [9].

In the foreseeable future, the FOWTs are considered more cost-effective than bottom-fixed offshore wind turbines in commercial operation with the significant technology advancements. However, several key factors for the design of FOWTs remain in need of more attention and effort. One of the key factors is the strong interaction between the wind turbine and floating platform. Unlike the bottom-fixed offshore wind turbines, the aerodynamics of FOWTs exhibits significant unsteady characteristics due to the floating platform motions excited by incident waves [10]. Additionally, the hydrodynamics of floating platforms is also significantly influenced by aerodynamic loads exerted on turbine rotors transmitted by the tower. Figure 1 shows the distributed flow field induced by the interactions between wind turbine and floating platform. It can be seen that the wind turbine is forced into and out of its wakes periodically due to the presence of the periodic pitch motion of floating platform, which will lead to the sophisticated characteristics of the aerodynamic power.

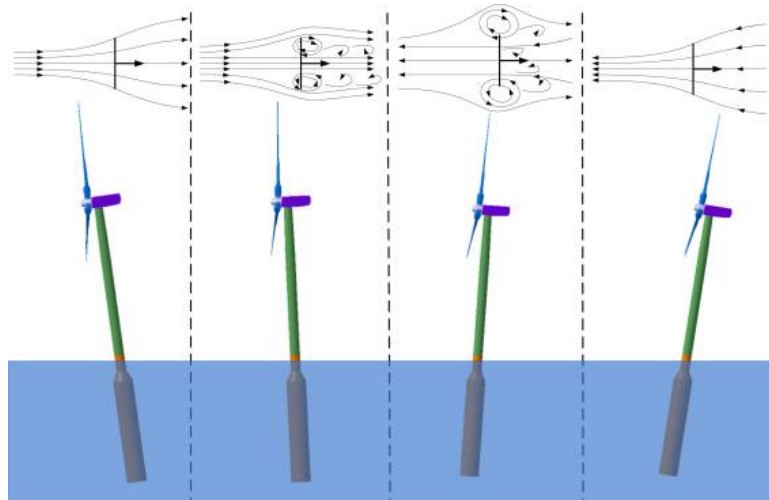

**Figure 1.** This disturbed flow field due to strong interactions of FOWT's aero-hydrodynamics [11].

In addition to the strong interactions between aerodynamics and hydrodynamics of FOWTs, another key factor is the aeroelastic due to the notable feature of the increasing rotor diameter. As previously mentioned, the LCOE of FOWTs is more expansive than that of the onshore wind turbines. The technology advancements in rated power for offshore wind turbines will bring a considerable cost reduction, approximately 8.5%, and is significantly better than the advancements in aerodynamics and floating platform design and installation [12]. Consequently, the development trend of FOWTs towards a large size, with the aim to be more commercially affordable, which makes the aeroelastic responses of the wind turbine blades very prominent. It is known that the aerodynamic performance of wind turbines is reduced by the aeroelastic responses. Additionally, the issues of fatigue loads and structural failure for wind turbine blades are more urgent, particularly under severe sea conditions. Therefore, with the great advancements of FOWTs towards large-

scale and deep-sea, the fully coupled aero-hydro-elastic performance requires a more systematic and comprehensive study to improve the reliability of FOWTs operated in combined wind-wave environments.

The research of the FOWTs is composed of three parts: the prototypes, the scale-down experiments and the numerical simulations, which are summarized in the previous literature reviews [13,14]. With the great advancements of high-performance computers (HPC), the numerical simulations for the FOWTs have gradually received a lot of attention. Some analysis codes have been developed and used [15,16]. Among those codes, the low-order modeling techniques incorporating the blade element momentum (BEM) theory [17] and potential flow (PF) models [5] are commonly employed in the design of FOWTs. Despite the low computational costs, those low-order modeling techniques cannot account for the viscous effects, which are important for calculating the aerodynamics of wind turbine blades, tower and hub, and the hydrodynamics of floating platform. Some correction models are required for BEM to guarantee the desired numerical results [18]. In other words, the ability of BEM on the predictions of aerodynamic loads needs further and comprehensive investigation [19], especially for FOWTs that the inflow wind conditions are more sophisticated due to the platform motions. In addition, the ability of potential-based models is limited on the accurate predictions of underlying flow mechanisms and its nonlinear dynamic characteristics, due to the fact that the flow separation around the platform cannot be captured [20]. In contrast, the above issues can be addressed by applying the high-fidelity computational fluid dynamics (CFD). Although the computational costs of CFD are usually expansive, it is expected that the costs will become more affordable due to the great advancements of HPC. Therefore, this study focuses on the applications of high-fidelity CFD in the studies of fully coupled performance of FOWTs.

The remainder of this study is constructed as follows: Section 2 describes the component-level studies of FOWTs, including aerodynamics, aeroelastic and hydrodynamics; in Section 3, the system-level studies consisting of simplified and fully coupled contents are summarized; the future recommendations are discussed in Section 4 and the conclusions are drawn in Section 5.

## 2. Component-Level Studies

### 2.1. Aerodynamics

In CFD-related simulations for wind turbine aerodynamics, two approaches exist: the actuator models, in which the wind turbine blades are represented by body force, and the direct modeling, in which the wind turbine blades are represented by computational and refined blade-resolved mesh. Representing wind turbine blades with actuator models has many advantages, for instance, avoiding solution of surface boundary layer of blades, saving the computational costs and easing the mesh generation. Generally, the actuator models are composed of three categories: actuator disk (AD), actuator line (AL) and actuator surface (AS), as shown in Figure 2. However, those actuator models are highly dependent on airfoil data, which is invalid for the design and development of novel wind turbine blades. By contrast, the direct blade-resolved modeling can easily handle this issue, and provide the rich and detailed flow field on the blade surface. Therefore, the two modeling methods are both widely used for the wind turbine aerodynamics.

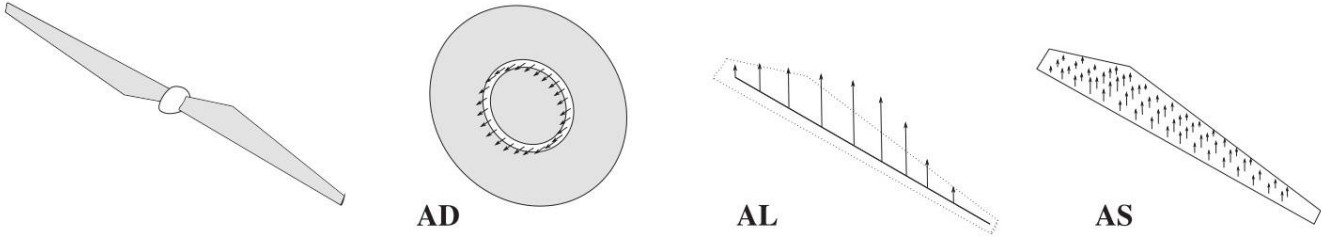

**Figure 2.** Three actuator models for wind turbine aerodynamics: AD, AL and AS [21].

### 2.1.1. Actuator Disk Model

For a uniformly loaded disk, the body force $f_{AD}$ exerts on disk surface can be expressed by:

$$f_{AD} = \frac{1}{2} V_{ref}^2 C_T \qquad (1)$$

where, $V_{ref}$ is the reference velocity, $C_T$ is the thrust coefficient. It is important to note that the determination of reference velocity $V_{ref}$ is critical to calculate the thrust coefficient $C_T$. Obviously, for an undisturbed uniform inflow, reference velocity $V_{ref}$ is evidently inflow velocity $U_\infty$. However, for the turbulent inflow situation, for instance, wakes of upstream wind turbines, this is not the case. This issue is usually addressed by introducing an iterative process based on the one-dimensional momentum theory [22], with it, the reference velocity $V_{ref}$ can be expressed by the function of axial induction $a$:

$$V_{ref} = V_{local}/(1-a) \qquad (2)$$

where, $V_{local}$ is the local velocity of disk surface. Additionally, the body force for a non-uniformly loaded disk is also related with the radial position, but the body force over the annulus remains constant (see Figure 2). The sectional coefficients of lift and drag are used to calculate the body force of wind turbine blades. In addition to the uniform and non-uniform loads of actuator disk, the rotational effects are also taken into account by introducing the tangential forces [23].

Once the body force calculated by the AD are obtained, as well as AL and AS, a projection procedure is used to distribute the body force smoothly in the flow field, with the purpose of reflecting the effects of wind turbine blades on flow field and eliminating the numerical singularity. The Gauss smooth function is widely adopted for the smooth projection procedure, and a desired and reasonable projection width is approximately two times the mesh resolution around turbine blades [24]. Another iterative smooth projection function is the discrete Delta function, where less cells are required to satisfy the conservation of forces and moments [25].

Revaz and Porté-Agel [26] performed a comprehensive evaluation of AD compared with an experimental test, the effects of several factors, i.e., projection parameter, model formulation, hub, tower and grid resolution on wind turbine aerodynamics and wakes were analyzed. Their findings revealed that projection parameter has a strong effect in rotor plane and a lesser effect in streamwise direction. The effects of model formulation are evaluated by comparing the numerical results between a simple AD which the loads applied on rotor are uniform and an advanced AD which the loads are non-uniform and determined by blade element theory. Both models exhibit accurate predictions for thrust, power and wind turbine far wakes, but the advanced AD performs better in near wakes. In addition, the velocity deficit is enhanced and aerodynamic predictions are decreased due to the presence of hub and tower, which is referred to as the tower shadow effects. The calculation results reach convergence when less cells are arranged along the radial direction of the rotor, by approximately 10 cells. Furthermore, Micallef et al. [27] assessed the ability of non-uniform loaded AD for the predictions of near wake expansion, they importantly noted that the AD gives a reasonable prediction of wake expansion in the radially outboard positions, but exhibits a poor situation for mid-board and inboard areas. Li et al. [28] proposed a numerical model by incorporating the AD and an extended k-epsilon turbulence model, as well as considering the effects of hub and tower. By comparing with experimental data, the proposed model shows slightly better than the standard AD, the prediction accuracy in the near wake region is improved and the overall prediction accuracy for the wind turbine wakes is certainly promoted.

Some researchers also focus on the improvements of the AD, with the aim of achieving easier operation or more reasonable predictions. Sørensen et al. [29] proposed an analytical model for calculating the body forces of the AD, and the advantages of this model are that the detailed knowledge of wind turbines are not required, but only the rated wind speed

and power capacity are needed. Conversely, Naderi et al. [30] employed an improved methodology for the AD to take into account all operational and geometrical characteristics of wind turbines, including airfoil type, angular velocity, twist, and chord distribution. Behrouzifar and Darbandi [31] developed an improved AD whereby the real thickness of wind turbine blades is considered, without the need to find the specific grid thickness of AD in the convergence tests, and thus the computational and temporal costs are reduced. Moreover, in order to consider the three-dimensional (3D) flow effects in radial direction, Amini et al. [32] introduced the 3D correction of aerodynamic coefficients into the AD. Compared to the original aerodynamic coefficients, the corrected 3D coefficients exhibit a better agreement with the experimental results.

### 2.1.2. Actuator Line Model

Different from the AD, the body force of wind turbine blades of AL is acting on the rotational lines [33]. Figure 3 shows the velocity vectors of a two-dimensional airfoil section. Obviously, the relative velocity $U_{rel}$ is determined by:

$$U_{rel} = \sqrt{U_z^2 + (\Omega r - U_\theta)^2} \tag{3}$$

where, $U_z$ and $U_\theta$ are the axial tangential velocity and tangential velocity of inflow wind, respectively, $\Omega$ is the rotor speed, and $r$ is the radial position. The body force $f_{AL}$ acting on actuator lines is calculated by the following equation:

$$f_{AL} = (L, D) = \frac{1}{2}\rho U_{rel}^2 c \left( C_L \vec{e_L} + C_D \vec{e_D} \right) \tag{4}$$

where, $L$ and $D$ are the life and drag at radial position of $r$, $\rho$ is air density, $c$ is chord length of the two-dimensional airfoil section, $C_L$ and $C_D$ are the lift and drag coefficients, $\vec{e_L}$ and $\vec{e_D}$ are unit vectors of lift and drag directions.

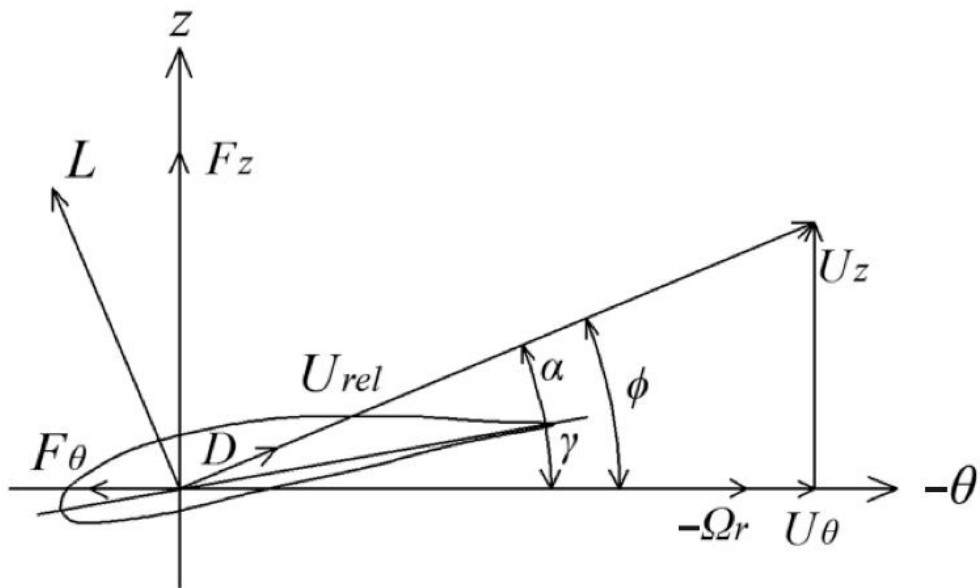

**Figure 3.** Velocity vectors of two-dimensional airfoil section [34].

Martinez-Tossas et al. [35] evaluated the ability of four Large Eddy Simulation (LES) analysis codes by using the AL to predict wind turbine aerodynamics and wakes. It was shown that the simulated results, i.e., velocity and force along the blades, velocity and Reynolds stress distributions in wakes across the four LES codes exhibit excellent agreement. Ravensbergen et al. [36] performed the wind turbine modeling based on the AL with a variational multiscale framework applied for turbulence modeling. The model

was firstly validated by NREL 5 MW wind turbine, and followed by the comparisons with two wind tunnel experiments and a wind field measurement. Figure 4 shows the vortex structures induced by the wind turbine, note that the wind turbine blades are modeled by AL, while the hub and tower are modeled by computational refined meshes. The blade tip vortices and hub vortices are obviously visualized, and the shed vortices of tower are mixed with wind turbine wake vortices when they travel downstream.

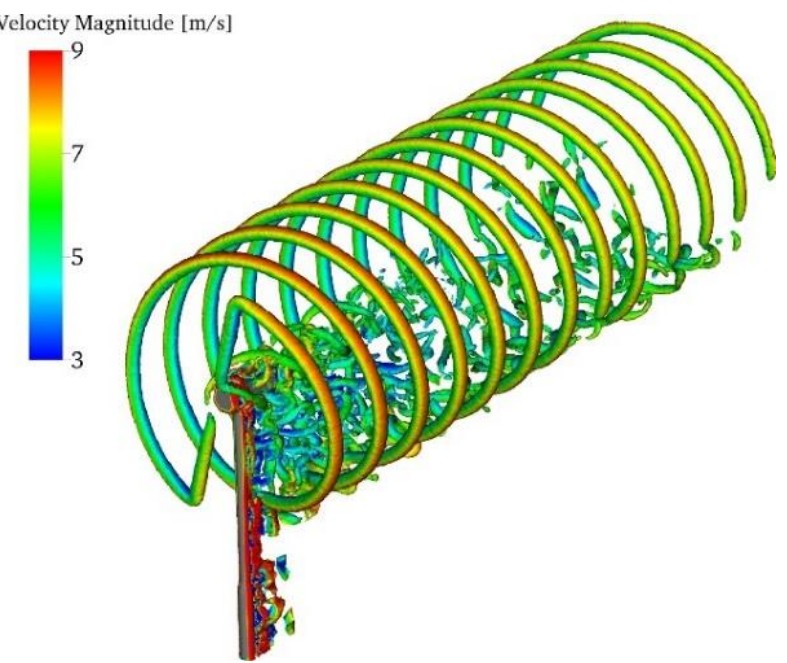

**Figure 4.** Vorticity isosurfaces ($Q = 15$ m$^2$/s$^2$) coloured by velocity magnitude for NREL Phase VI wind turbine [36]. Reproduced with permission from Ravensbergen et al., Computers & Fluids; published by Elsevier, 2020.

In addition to the stand-alone wind turbine modeling, the AL also shows its excellent capability for the modeling of multiple wind turbines. By conducting the simulations of two aligned NREL 5 MW wind turbines with the AL, Yu et al. [34] pointed out the non-negligible influence of upstream wind turbine wakes on the downstream wind turbine aerodynamics, and a reasonable spacing of two aligned wind turbines is approximately 7D (D is the rotor diameter). Onel and Tuncer [37] also noticed the significant effects of upstream wind turbine wakes on downstream wind turbine aerodynamics by presenting simulation cases of two aligned wind turbines modeled by the AL. The downstream wind turbine suffers a severe power loss of 86% under non-turbulent inflow, whereas this situation is alleviated for turbulent inflow that the power loss is reduced to 64%. To provide the potential insight for wind farm control strategy and power improvement, Draper et al. [38] performed the numerical simulations for a stand-along wind turbine considering torque controller and three wind turbines with two yaw settings, the numerical results show good agreement with wind tunnel experiments.

Similar to the AD, some efforts are dedicated to improving the ability of AL for the wind turbine modeling and obtaining the more desired results. Jha et al. [39] proposed an advanced AL by introducing an actuator curve embedding (ACE) concept to eliminate some inconsistencies in the body force projection process. Several examples are used to assess and validate the behaviors of ACE concept and the good results are presented. Xie [40] modified the standard AL by replacing conventional spatial-averaged velocity sampling with an Lagrangian-averaged velocity sampling, as well as replacing Gauss function by a piecewise function for the body force projection. The validated example of NREL 5 MW wind turbine showed that the proposed model can effectively reduce spurious numerical oscillations. Additionally, an advanced AL considering the corrections of tip

and root loss, as well as 3D airfoil delayed stall, was conducted by Xue et al. [41] to better capture the aerodynamics and wake characteristics of wind turbines. The hub and tower are also modeled by the AL, and obviously, the advanced AL shows better agreement with experiments, compared to the standard AL.

### 2.1.3. Actuator Surface Model

The AL was extended to AS by Shen et al. [42,43], in which the wind turbine blades are represented by planar surfaces. Compared to the AL, the AS requires more detailed airfoil data, including the distributions of pressure and skin-friction on the airfoil surface. The body force $f_{AS}$ acting on actuator faces is expressed by:

$$f_{AL}(\xi) = (L, D) = \frac{1}{2}\rho U_{rel}^2 c \left( C_L \vec{e_L} + C_D \vec{e_D} \right) F_{dist}(\xi) \tag{5}$$

where, $\xi$ is a distance factor of airfoil chord, $F_{dist}(\xi)$ is determined by the fitting empirical functions of chordwise pressure distributions, which can be obtained by using Xfoil, a highly accurate tool to compute pressure-friction and skin-friction profiles on airfoils [21].

Sibuet et al. [44] applied the AS to the aerodynamic predictions of wind turbines, and validated its ability for aerodynamic modeling against experiment measurements and other numerical methods. They also claimed its applicability in modeling wind turbine aerodynamic wakes. To further consider more details of wind turbine blade geometry and the effects of nacelle, Yang and Sotiropoulos [45] developed a new AS for wind turbine modeling. The new model was firstly validated by a stand-alone nacelle with the wall-resolved LES results, and then followed by lab-scale wind turbine and a hydrokinetic turbine. These validate examples demonstrated that the new AS has the ability to accurately capture the wake characteristics of wind turbines. They also presented a systematic study for wake characteristics of a utility-scale wind turbine under various operational conditions [46]. The results illustrated that the wake meandering, which is responsible for downstream wind turbine operations, is attributed by the coexistence of large-scale atmospheric turbulence structures and shear instabilities of wind turbine wakes. Foti et al. [47] investigated the flow over a model wind turbine by using the above proposed AS for wind turbine blades and nacelle, and compared the results with wind tunnel experiments. Apart from the model wind turbine, they also examined the effects of nacelles on aerodynamics and wakes of a large wind farm by the novel AS with and without the nacelles [48]. It was found that the nacelles have a significant influence on wind turbine power fluctuation and turbulence intensity in wakes, as well as the wake meandering. Moreover, the AS has been employed as a validation and verification tool. Except for the extensively validated time-averaged quantities, Li and Yang [49] also assessed the ability of AD for the dynamic predictions of wind turbine wakes by the well-validated AS.

### 2.1.4. Direct Modeling

The wind turbine blades are fully resolved by fine grids called direct modeling, which is a high-fidelity method in CFD simulations and significantly different with the afore-mentioned actuator models. There are no requirements for the direct modeling method to pre-obtain the lift and drag coefficients of airfoil sections, and the 3D effects of radial flow are automatically considered. Compared to the actuator models, the direct modeling method can capture more flow details on blade surfaces and show its excellent ability for the design and development of novel wind turbine blades.

Sutrisno et al. [50] importantly noted the unstructured mesh of wind turbine blades probably can lead to significant error results for wind turbine aerodynamic characteristics, whereas more accurate numerical results are obtained by the structured mesh. Consequently, in order to guarantee excellent mesh quality and perform better numerical simulations of flow pass the wind turbine rotor and wakes, the structured multiblock hexahedral mesh was used by Regodeseve et al. [51] to represent a wind turbine, as shown in Figure 5. In addition, the computational domain is large enough to consider the development of

wind turbine wakes, and the rotor rotation is simulated by sliding mesh technique. A comprehensive comparison with the experiment measurements was concluded, and as expected, the CFD predictions shows a good consistency. Wang et al. [52] also applied the sliding mesh in commercial code STAR-CCM+ to investigate the effects of different tilt angles on the wind turbine aerodynamics. It was found that the tilt angle has significant impacts on the aerodynamic performance of wind turbines, and the best tilt angle is of 4°. Zahle et al. [53] performed investigations for the interactions between rotor and tower by using the overset mesh technique. The computations are in accordance with experimental data and successfully reproduce the strong rotor-tower interactions characterized by increasing transient loads on wind turbine blades.

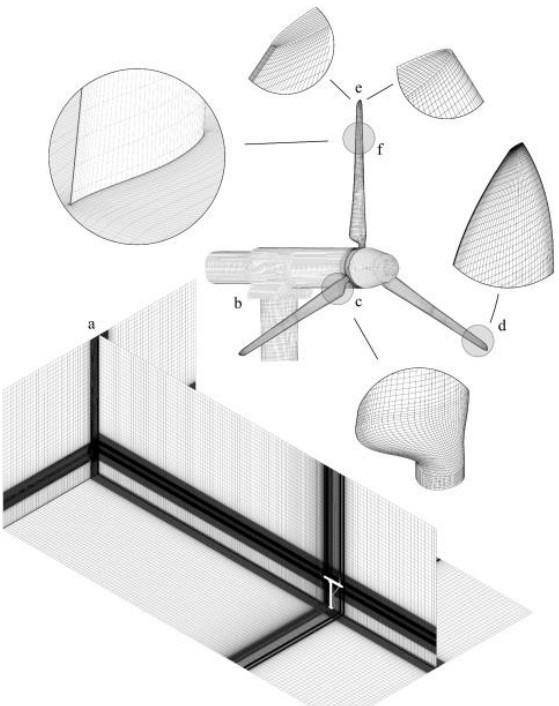

**Figure 5.** Computational domain: (**a**) structured multiblock hexahedral mesh; (**b**) complete wind turbine (blades, hub, nacelle, and tower); (**c**) root of the blade; (**d**) tip of the blade; (**e**) trailing and leading edge at the tip; (**f**) boundary layer [51]. Reproduced with permission from Regodeseve et al., Energy; published by Elsevier, 2020.

Li et al. [54] also examined the wind turbine aerodynamics using overset mesh technique, and two turbulence models, Reynolds-Averaged Navier-Stokes (RANS) equations and Detached Eddy Simulation (DES), were used for a comparison purpose. Although the time-averaged forces and moments of the two turbulence models exhibit a little discrepancy, a significant increase in transient responses is predicted by DES. Purohit [55] evaluated the accuracy of LES and unsteady RANS (uRANS) turbulence models on wind turbine rotor and wake aerodynamics. They concluded that the LES can better capture the flow separation than the uRANS when the wind turbine is under the stall operational condition. Sedano et al. [56] assessed the ability of Improved and Delayed Detached Eddy Simulation (IDDES) model on a model wind turbine aerodynamics. Compared to the DDES model, the IDDES shows a trend closer to the experimental data, with almost the same computational costs. Additionally, in order to investigate the distinctions between model-scale and full-scale wind turbines, Pinto et al. [57] simulated the wind turbine aerodynamic behaviors in full and mode scale based on a commercial code. The power and thrust coefficients were compared with experiments for the model scale and other numerical codes for the full scale. It was concluded that the present CFD predictions are generally in line with available numerical results, but showed a good match with experimental data.

Due to the experimental measurements not being easily available, the high-fidelity direct modeling for wind turbine aerodynamics is adopted for the validation of other relative lower-fidelity numerical methods. Bangga [58] validated a new code, namely B-GO, using BEM for wind turbine aerodynamics against the blade-resolved CFD computations. The comparisons demonstrated that the developed code has an excellent ability for the predictions of wind turbine aerodynamics. Carreno et al. [59] compared the BEM and free vortex wake (FVM) with 3D CFD simulations for the aerodynamics of a wind turbine with two blades. Their findings proved that the radial local thrust on turbine blades predicted by FVM is more accurate than that of the BEM.

## 2.2. Aeroelasticity

For the aeroelastic analysis of wind turbines, both the aerodynamic models to determine the forces exert on turbine blades and the structure models to determine the structure dynamics are required. Figure 6 shows the commonly used aerodynamic and structure models for aeroelastic modeling of wind turbine blades. Regarding the aerodynamic models, we are more focused on the high-fidelity modeling for wind turbine aerodynamics, for instance, the CFD-related actuator models and the direct modeling that have been described and discussed in Section 2.1. The structure models used in wind turbine aeroelasticity are categorized to two groups, 3D finite element method (FEM) and 1D equilibrium beam model (EBM). Undoubtedly, the 3D FEM is capable of providing more accurate structure deformations by using the shell or solid elements to discretize the composite blades, but we suffer expensive computational costs in doing so. In the 1D EBM, the blades are discretized to a series of beam elements by three discretization methods, specifically, model approach, multi-body dynamics (MBD) and 1D FEM. In order to construct the 1D EBM for the aeroelastic analysis, the cross-sectional analysis model is employed to obtain the cross–sectional properties of blades, for instance, the cross-sectional stiffness and radial distribution of blade mass. In this study, we are more inclined to the introduction and discussion of the three discretization methods, more theoretical details about the cross-sectional analysis model can refer to [60].

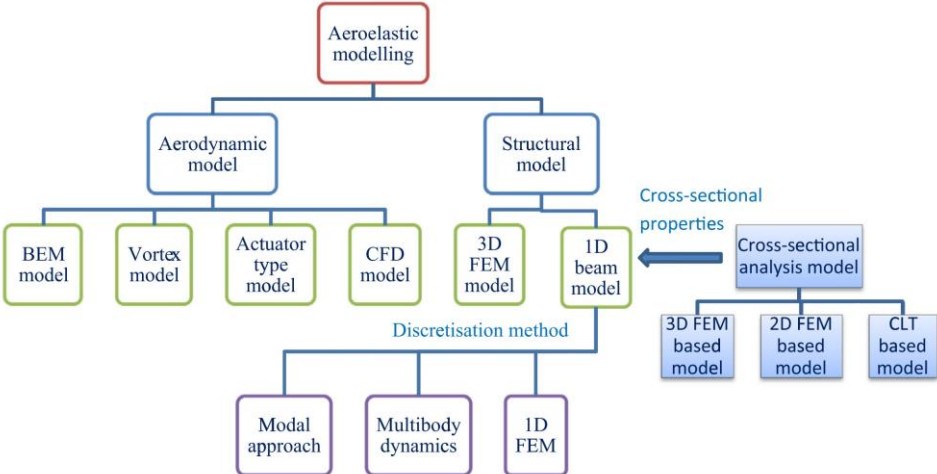

**Figure 6.** The commonly used aerodynamic and structure models for aeroelastic modeling of wind turbine blades [60]. Reproduced with permission from Wang et al., Renewable and Sustainable Energy Reviews; published by Elsevier, 2016.

### 2.2.1. One-Dimensional Equilibrium Beam Model

The 1D EBM can be used efficiently to model the structures of wind turbine blades, owing to the slender characteristics of wind turbine blades. Roughly, the 1D EBM consists of two categories, linear beam models and nonlinear beam models. The Euler-Bernoulli beam model [61] and Timoshenko beam model [62] are the two widely used linear beam models. Compared to the Euler-Bernoulli beam model, the cross-sectional shear effects

of turbine blades are accounted for the in the Timoshenko beam model. However, for the slender structures of turbine blades, the difference of the calculated deformations by the two beam models shows little distinction. The linear beam models have an assumption of small deformations, which is unreasonable and invalid for the wind turbines experience the situations of large deformations, such as the extreme wind conditions. Consequently, the nonlinear models have been proposed to consider the geometrically nonlinear characteristics of wind turbine blades and address the issue of large deformations. One of the well-known nonlinear beam models is the geometrically exact beam theory (GEBT) [63]. Except the rigid cross-sectional assumption, the GEBT does not contain other assumptions and the displacement-strain relationship can be used for significant displacement and rotation, which is suitable for the analysis of geometrically nonlinear beams.

As mentioned in Figure 6, three discretization methods can be used to discretize the beam model into a series of beam elements, i.e., modal approach, MBD and 1D FEM. The modal approach is computationally efficient by representing the deformations of turbine blades based on the superposition of modes. Due to the linear superposition assumption of the modes, the modal approach is not appropriate for the large and nonlinear deformations of wind turbine blades, and the solution accuracy depends on the prescribed modal shapes. Additionally, the flutter analysis is excluded in modal approach because of the reduced degrees-of-freedom of beam elements. The MBD discretizes the blade structure into a series of independent beam elements, each of which is constrained by force or motion relationships. Compared to modal approach, the MBD suffers more computational costs, but the flutter analysis is considered and thus more accurate results are guaranteed [64]. The 1D FEM discretizes the blade into a series of beam elements that are connected through the internal force and displacement at the nodes, and solves the deformation in combination with boundary conditions. Among the three discretization methods, 1D FEM can more accurately describe blade deformations and a little more computational costs are required compared to the MBD. Consequently, the 1D FEM has been widely used in the aeroelastic analysis of wind turbine blades [65].

Gözcü and Verelst [66] studied the effects of fidelity levels of structure models on wind turbine load responses using the MBD and Timoshenko beam. The results showed that the load responses converge quickly to that of highest fidelity structure model with the increasing bodies, and indicated the significant effects of geometric nonlinearity for wind turbine blades. However, the low-fidelity BEM model was employed in the above study to calculate the wind turbine aerodynamics. For the certain and normal conditions, the low-fidelity BEM model combined with 1D structure model can capture correct aeroelastic responses for small wind turbines. However, for the situations of large wind turbines with significant blade tip deformations, the ability validation of those engineering models is of great needed and the high-fidelity models are suggested. Sayed et al. [67] evaluated the impacts of various fidelity levels of aerodynamic models on the aeroelasticity of DTU 10 MW wind turbine. The BEM model and CFD method incorporated with MBD solver SIMPACK were used, and Timoshenko beam was employed to represent the structures of rotor blades. It was found that the power and thrust predicted by the BEM-related aeroelastic model are smaller than that of CFD-related model. In addition, the effects of aerodynamic model fidelity are more pronounced for higher wind speeds. Li et al. [68] also presented a high-fidelity aero-servo-elastic framework for wind turbines by incorporating the CFD overset mesh technique and the MBD. The interactions between turbulence inflow, aeroelastic responses of turbine blades and the drivetrain dynamics were investigated. Similarly, Guma et al. [69] conducted a high-fidelity model for the aeroelastic responses of a 2 MW NM80 turbine subjected to turbulence inflow conditions by incorporating the CFD method for aerodynamics and the MBD for structure deformations. Different CFD modeling approaches with increasing complexity were performed to investigate its effects on wind turbine aeroelasticity. Grinderslev et al. [70] performed the aeroelastic simulations for a 2.3 MW wind turbine using CFD-based model with the MBD. Three inflow conditions, including the axisymmetric flow, the highly sheared flow and highly yawed-sheared flow,

were employed. Their results were compared to BEM-MBD model and validated through field experiments. According to their analysis, the further validations of BEM-based model for complex flows are recommended through high-fidelity model.

Except the MBD-related models for aeroelastic analysis of wind turbine blades, some researchers have also focused on the 1D FEM-based models. Sayed et al. [71] conducted a high-fidelity CFD-CSD (computational structure dynamics) aeroelastic analysis for the DTU 10 MW wind turbine. The CFD solver FLOWer was utilized to calculate the aerodynamic loads and the CSD solver Carat++ was employed to predict the structure deformations based on the 1D FEM. Figure 7 shows the CFD surface mesh and CSD structure mesh. The numerical results reflected that the aerodynamic loads are reduced due to the aeroelastic deformations of turbine blades. Additionally, to investigate the coupling effects between aerodynamic models and structure models on the aeroelastic responses, the proposed CFD-CSD framework was compared with the CFD-MBD and BEM-MBD frameworks. Three years later, they presented a validated and comprehensive aeroelastic analysis based on the previous proposed framework [72]. The rotor power and thrust were enhanced due to the contribution of radial force induced by the edgewise deformations. The tower effects were discussed, in which the deformations and forces were decreased when blades passed the tower. Dose et al. [73] developed an aeroelastic tool for wind turbines by coupling CFD code OpenFOAM with the inhouse developed structure code BeamFOAM. The blade structures were represented by GEMT to account for the large deformations and discretized by finite elements. Their goal was to investigate how the blade deformations affect the wind turbine aerodynamic responses, i.e., rotor power, thrust, and cross-sectional forces. The results also demonstrated the same conclusion that the significant effects of blade aeroelasticity on aerodynamic response, especially for the yawed inflow conditions.

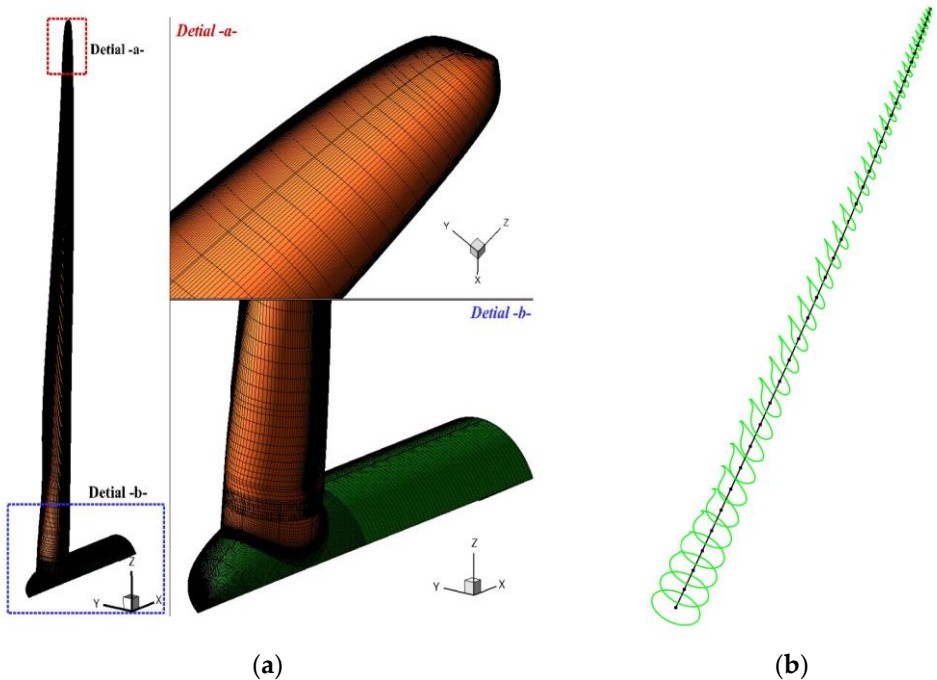

(**a**)                                         (**b**)

**Figure 7.** CFD-CSD mesh: (**a**) CFD surface mesh; (**b**) CSD beam finite elements (black lines), and the green sections are the predefined cross sections [71].

Due to the reasonable accuracy of the AL for wind turbine aerodynamics and its mature property, the turbine blade is regarded as a line which perfectly match the theory of the EBM, the AL-related aeroelastic analysis model has been developed and utilized recently for the aeroelasticity of wind turbines, namely the elastic actuator line (EAL). By combining the AL and a modal approach structure solver, Della et al. [74] presented an aeroelastic model for wind turbines. The distinction of one-way and two-way loose

coupling methods between the fluid and structure solvers was discussed. Overpredicted results are found for rotor power and blade deformations when using the one-way coupling method, indicating the necessity of utilizing the two-way coupling method. Yu et al. [75] proposed a new EAL analysis model by combining the AL and 1D FEM to predict the aeroelastic responses of wind turbines rapidly. The influence of blade deformations and tower effects on wind turbine aerodynamic performance were examined and discussed. Moreover, the results indicated that the above effects on downstream wind turbine are more significant. Furthermore, Ma et al. [76] developed an aeroelastic framework for wind turbines based on the combination between the AL and the nonlinear beam model with finite elements. The study focused on the aeroelastic wake behaviors of NREL 5 MW wind turbine and found that the vorticity and velocity recovery in far wakes are underpredicted due to the absence of blade deformations. Meng et al. [77,78] examined the fatigue loads on downstream wind turbines by the EAL model. The complex atmospheric inflow was considered, and two in-line wind turbines and a wind farm with nine wind turbines were performed. A significant enhanced fatigue damage for downstream wind turbines was discovered, compared to that of upstream wind turbines.

### 2.2.2. Three-Dimensional Finite Element Method

The required accuracy of aeroelastic responses for wind turbines has a great impact on the selected structure models. To pursue more precise and detailed results of wind turbines, the 3D nonlinear FEM is a more suitable option. The 3D FEM structure models are broadly divided into two categories [79], the shell element model and the solid element model, as shown in Figure 8. The shell element model is usually used to predict the cross-sectional deformations of turbine blades. However, if the detailed stress or damage estimation of turbine blades are desired, the solid element model is commonly adopted.

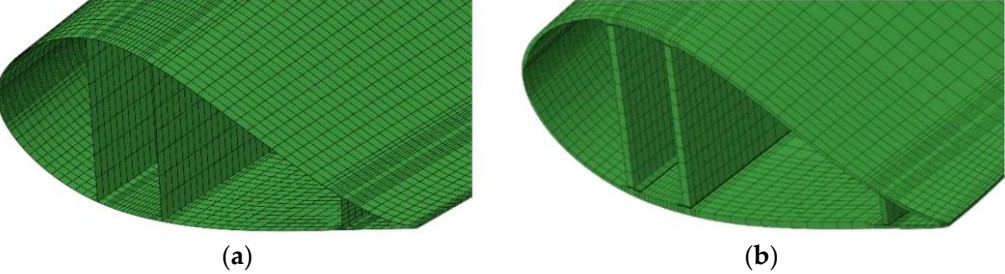

|              (**a**)              |              (**b**)              |

**Figure 8.** The 3D FEM structure models: (**a**) shell element model; (**b**) solid element model [80].

To handle the aeroelastic design of a high power wind turbine, Tojo and Marta [81] developed a fluid-structure interaction (FSI) solver in OpenFOAM. The blade structures were represented by solid hexahedral elements and the blade rotation was addressed by a single rotating framework. They noted that the aeroelastic responses of wind turbines is definitely a problem and should be considered when the turbine size increases. Zhangaskanov et al. [82] performed an investigation of the aeroelastic performance of NREL Phase VI wind turbine based on a FSI solver, which is also implemented in OpenFOAM. The turbine blade structures were represented by the solid element model. The simulation results showed a good agreement with experiments, outlining the accurate prediction ability of the high-fidelity CFD-CSD framework for aeroelastic analysis of wind turbines. Peeters et al. [80] studied the structure deformations of a wind turbine blade under static loads, both the shell element model and the solid element model were used to model the wind turbine blade. Compared to the experiment measurements, the differences of the two models were minor, reflecting the sufficient prediction accuracy of blade structure deformations under the specific load case for the shell element model.

Similar with the comparison between the solid and shell element models, Guma et al. [83] conducted a comparison between the beam model and the shell element model for the aeroelastic analysis of a small wind turbine by applying the coupling between the CFD

solver FLOWer and the FEM solver Kratos. The results indicated that the beam model is sufficient to predict the blade deformations for a wind turbine blade under uniform inflow conditions. However, when the complexity of simulation cases increased, i.e., the full wind turbine under turbulent inflow conditions, the shell element model is required to give more reasonable results. Shkara et al. [84] investigated the interactions between elastic blades and tower by combining a CFD solver and a FEM solver. The structure of blades and tower were discretized through shell element model. It was concluded that the rotor azimuthal position has a significant effect on the structure dynamic responses of tower. Additionally, they also performed the same simulations using the BEM-MBD aeroelastic framework, and higher flapwise and edgewise deformations of turbine blades were predicted. Apart from the rotor-tower interaction, Santo et al. [85] also presented the effects of tilt angle and yaw misalignment on the aeroelastic performance of wind turbines. They observed that the blade deformations are significantly affected by tilt angle due to its contribution to gravity, and a reduction in blade axial displacement is observed.

### 2.3. Hydrodynamics

Four types of floating platforms are used to support the wind turbines: spar buoy, semi-submersible, tension leg and barge concepts, as illustrated in Figure 9. The spar buoy platform adopts a vertical cylindrical structure characterized by simpler design, higher stability and lower wave-induced motions. The drawbacks of the spar buoy platform are the difficulty in transportation and installation due to the deep draft, as well as the high fatigue loads for tower. The larger floating structure and wider water-line area are adopted by semi-submersible platform to maintain the structure stability. Compared to the spar buoy platform, the operational water depth of semi-submersible platform is not limited because of its low draft and towing convenience. But the complex structures of the buoyancy floater, pontoon and supporting brace are difficult to design and manufacture [5]. With respect to the tension leg platform, the buoyancy of the floater is balanced with the gravity and the tension forces of three or four vertical mooring cables connected from the floater to the seabed. The tension leg platform has the advantages of simpler structure, lower fatigue loads and wave-induced motions, whereas the particularly rigorous design of tensional mooring cables is recommended. The main structure of barge platform is a square platform with shallow draft and wider waterline area. The advantages of this kind of platform are less complexity and lower costs, however, it is sensitive to external environment and not suitable for extreme sea conditions.

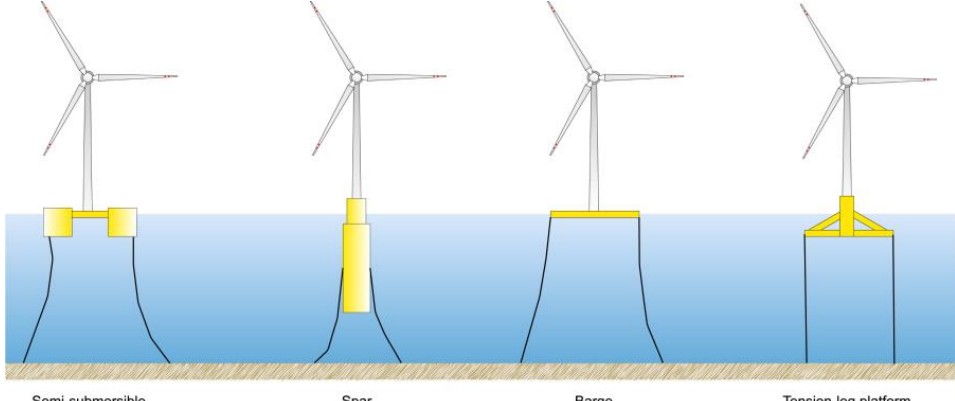

**Figure 9.** The supporting platforms of FOWTs [86]. Reproduced with permission from Micallef et al., Renewable and Sustainable Energy Reviews; published by Elsevier, 2021.

Due to the mature design of floating oil and gas platforms, three commonly used approaches, Morison equation (ME), PF method and CFD method are employed for the hydrodynamic performance of FOWTs. The ME cannot account for the effects of floating platforms on the incident wave field [13], which limits its application in the hydrody-

namics of floating platforms. Additionally, the ability of the PF method for predicting the hydrodynamic responses of floating platforms dominated by viscous force remains questionable, i.e., the complex flow issues of wave climbing and slamming. Compared to the ME and PF methods, more physical flow mechanisms such as fluid viscosity, wave diffraction, radiation, wave climbing and slamming can be captured by CFD method based on the solution of Navier-Stokes equation. Generally, hydrodynamic studies with CFD method for these platforms are more focus on loads and motion responses induced by incident waves and vortices. Figure 10 shows the definition of six degrees-of-freedom (6-DOF) for FOWTs, including three translational components (surge, sway, heave) and three rotational components (roll, pitch, yaw). Robertson et al. [87] noted that the surge and pitch frequencies of floating platforms are usually underestimated due to the absence of fluid viscosity and other related physical quantities. Consequently, the predictions of hydrodynamic performance of floating platforms by CFD method are more accurate and detailed information of flow field is available.

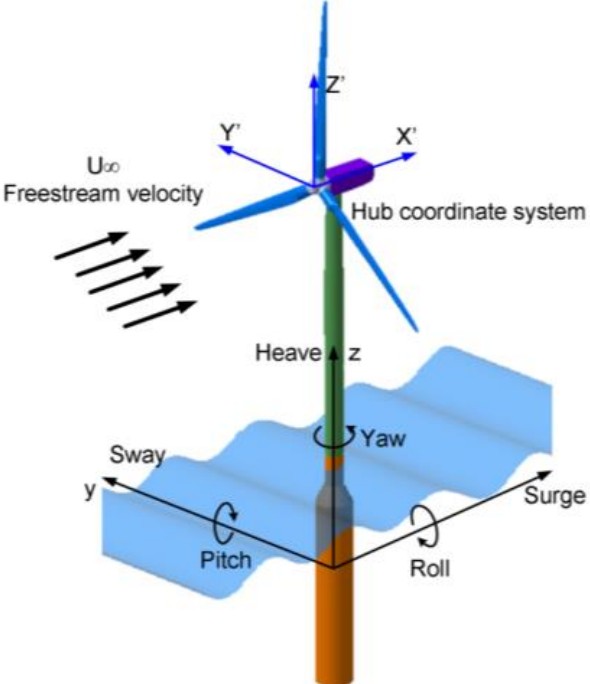

**Figure 10.** Definition of the platform motions for an offshore floating wind turbine [88]. Reproduced with permission from Tran et al., Journal of Mechanical Science and Technology; published by Springer Nature, 2015.

In hydrodynamic modeling, the commonly used mesh generation technologies for CFD are dynamic mesh technique [89] and overset mesh technique [90]. Each method has its own advantages and specific limitations. For example, the accuracy and numerical stability of dynamic mesh technique are affected by the mesh quality, which is poor for the situations of significant motions of floating platforms. In contrast, the mesh quality of floating platforms with significant motions is guaranteed by the overset mesh technique, whereas the mature experience of researchers in mesh generation and more computational costs are required.

The extreme waves caused by typhoon have a huge impact on the dynamic responses on floating platforms, which is characterized by structure instability and incompleteness [91]. Consequently, the mooring system is required to limit the dynamic responses and guarantee the structure stability of FOWTs. The numerical analysis methods of mooring systems can be roughly divided into static method, quasi-static method and dynamic method [92]. The static method considers only the constant loads, such as gravity, buoyancy, steady current and wind, as well as mean wave-drift forces. The quasi-static method is

proposed since there is no absolute static state. It assumes that within a given time step, the motion of the system is uniform and linear between two static positions, and the loads of the system are constant. Quallen et al. [93] combined the quasi-static claw mooring model with the CFD solver CFD-Ship Iowa v4.5 to perform the free decay of a OC3 spar buoy platform. The dynamic method is used to obtain more accurate results of loads and significant displacements, including finite difference model (FDM), finite segment model (FSM), finite element model (FEM) and lumped mass model (LMM) [94]. The difference between FDM and FEM is the form of governing equation. Specifically, the FDM is based on a differential form, while the latter is based on an integral form. Moreover, if the resolution of lumped mass method is sufficient, it will exhibit the same solution as FDM and FEM [95]. For the analysis methods of mooring system, the quasi-statics model has high computational efficiency but with a limited accuracy that cannot reasonably address the problems of dynamic coupling and nonlinear deformations of mooring cables. In contrast, the dynamic analysis model has a higher accuracy and wider applications, but suffers more computational costs.

### 2.3.1. Spar Buoy Platform

The free decay of a spar buoy FOWT was performed by Beyer et al. [96] through the coupling between the multi-body system (MBS) software SIMPACK and the CFD. The pressure distribution on the platform surface caused by pitch motion was analyzed based on the body-fitted mesh. Quallen and Xing [97] performed a full-system, two-phase CFD simulation of a OC3 spar buoy FOWT with the crowfoot mooring system. The predicted maximum values of velocities of surge and pitch are decreased by 32.7% and 31.4%, respectively, compared to that of FAST, arising from the large mooring tension forces to keep the platform stability. Liu and Yu [98] investigated the dynamic behaviors of a spar buoy FOWT under JONSWAP-based wave group conditions generated by the envelope amplitude approach. The surge and pitch motions increase slightly with the increase in wave group, whereas the heave motion increases significantly. In addition, the low-frequency resonance response is easily excited by the wave group. Nematbakhsh et al. [99] conducted nonlinear simulations of a spar buoy FOWT under extreme sea conditions using the STAR CCM+. They drew a conclusion that higher aspect ratio spars will cause lower mean surge and pitch responses, but also may lead to a nonlinear trend in the standard deviations in pitch and heave.

With the incoming currents, the vortex-induced loads will cause the 6-DOF motions which called vortex-induced motions (VIM). The application of helical strake is a commonly used method to reduce the VIM of spar buoy platform. Zhao et al. [100] discussed the effectiveness of helical strake on suppressing VIM of spar buoy platform based on an in-house CFD code naoe-FOAM-SJTU [101]. Similar with the helical strake, the dynamic responses of spar buoy platform are also reduced by heave plate. Wang et al. [102] simulated the forced oscillation of heave plates with different shapes and holes using dynamic mesh technique. It was indicated that the damping coefficient of fractal plate is obviously larger than that of regular ones, besides, the plate with nonlinear holes exhibits better hydrodynamic performance. Subbulakshmi and Sundaravadivelu [103] performed the CFD simulations of a spar buoy platform with single and two heave plates and compared with experiments. Zheng et al. [104] investigated the dynamic responses of a OC3 spar buoy FOWT with the heave plate. They drew a conclusion that the amplitudes of heave and yaw are decreased by 52.259% and 46.836% due to the damping effects of the heave plate.

### 2.3.2. Semi-Submersible Platform

Semi-submersible platform is a burning research topic in recent years. Dunbar et al. [105] developed a CFD solver based on OpenFOAM including the 6-DOF equation for floating platforms. The developed CFD solver was applied to free decay tests of heave and pitch of a semi-submersible FOWT. They found that there is greater discrepancy between the predicted results of CFD and FAST in first few periods of heave decay, which is at-

tributed to the viscous effects. Cheng et al. [106] performed the free decay tests of a OC4 semi-submersible platform based on an in-house code naoe-FOAM-SJTU. The restraint of mooring cables was taken into account. The results of free decay simulations of platform surge and pitch are shown in Figure 11. It was indicated that the effects of fluid viscous on platform hydrodynamics can be reflected using the CFD method. Huang et al. [89] also performed the free decay tests of the OC4 semi-submersible platform, and the numerical results showed a good agreement with experimental data.

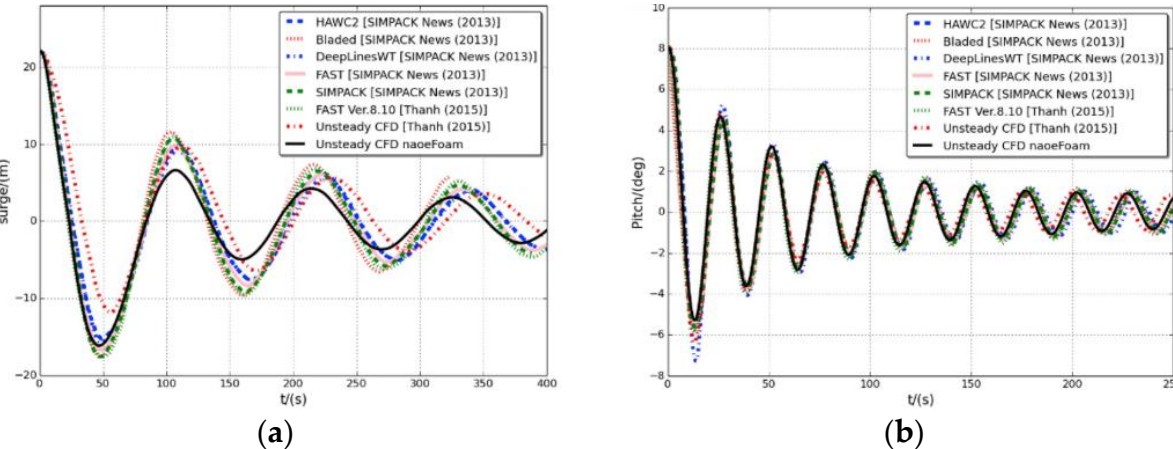

**Figure 11.** The free decay tests of a OC4 semi-submersible platform: (**a**) surge free decay; (**b**) pitch free decay [106]. Reproduced with permission from Cheng et al., Ocean Engineering; published by Elsevier, 2019.

The hydrodynamic loads of a OC4 semi-submersible platform were predicted by Benitz et al. [107] using the OpenFOAM and compared with experimental measurements. The shadowing effects and transverse forces from vortex were successfully captured under the current-only and wave-only conditions. Zhao et al. [100] studied the heave and pitch responses of a parked wind turbine mounted on the OC4 semi-submersible platform under various wind speeds. Tran and Kim [90], respectively, used the CFD method with overset mesh technique and the PF method to simulate the hydrodynamic responses of a OC4 semi-submersible platform under regular waves. The comparisons of the effects of various turbulence models on numerical results were conducted with the baseline reference of experimental data. Rivera-Arreba et al. [108] also used the CFD and the second-order PF methods to examined the hydrodynamics of a OC5 semi-submersible platform. Their main conclusion was that the PF method underestimates the amplitude of platform heave response by 40% compared to that of CFD method.

To investigate the effects of nonlinear waves on floater hydrodynamics, Liu and Hu [109] conducted CFD simulations of a semi-submersible platform under extreme sea conditions where the strongly nonlinear waves and wind loads are both considered. Wang et al. [110] focused on the uncertainty assessment of CFD predictions for the OC6 semi-submersible platform under nonlinear incident waves. Compared to second-order PF theory, the higher difference frequency wave excitations are captured by the CFD method, particularly for the surge response. In order to predict the nonlinear wave loads on a semi-submersible FOWT under regular and irregular waves, Li and Bachynski [111] developed a CFD model and an engineering model based on PF theory. They outlined that more accurate estimations of high-order wave forces can be captured by the CFD model, and the difference of wave loads between the CFD predictions and experiments is less than 10% under the regular wave conditions. By using two CFD analysis tools ReFRESCO and ComFlow, Bandringa and Helder [112] studied the deterministic breaking wave impact on a semi-submersible platform.

Nowadays, the research of VIM for semi-submersible platforms is a burning issue and many efforts have been dedicated to it. Tan et al. [113] investigated the VIM of a

multi-column floating platform with 3-DOF. The results revealed that the current heading, column aspect ratio, the sharp and rounded corners have significant effect on the vortex, and subsequently, affect the VIM of the platform. Kim et al. [114] examined the large period vortex-induced motion of a deep-draft paired-column semi-submersible platform at different heading angles based on the DDES turbulence model. Compared to the experimental measurements, the CFD predictions showed a satisfied accuracy. Zhao et al. [115,116] conducted the VIM of a semi-submersible platform with different square columns based on a well-validated in-housed code VIM-FOAM-SJTU. The results showed that the pontoon can effectively reduce the VIM of the platform.

### 2.3.3. Tension Leg Platform

Compared to spar buoy and semi-submersible platforms, the CFD studies of the tension leg platform are relatively few. Nematbakhsh et al. [117] found that the tension leg platform exhibits an obvious nonlinear characteristic under waves with large height. Compared with the linear assumption, lower platform surge is captured by CFD method due to better consideration of nonlinear effects. Dai et al. [118] conducted CFD simulations of a tension leg platform to calculate the global drag force. It was found that the discrepancy between CFD predictions and experimental measurements is less than 10%. Moreover, the mean drag coefficient is significantly contributed by the front column and pontoon. Nam et al. [119] performed the experiment and CFD simulation to study the relative wave elevation responses of a tension leg platform. A series of tests focused on the modeling of drift force and the occurrence of wave-in-deck were presented, and the nonlinear characteristics of relative wave elevation were analyzed.

### 2.3.4. Barge Platform

The dynamics responses of a barge platform under regular waves were simulated by Beyer et al. [120] using the coupled MBS-CFD method. The results of wave elevation, platform motion and mooring tension forces were found in good agreement with the experiment. The moonpool has been designed and installed on barge floating platform, with the benefit of improved seakeeping, reduced construction costs and increased potential multi-purpose applications [121]. Kristiansen et al. [122] simulated the moonpool resonance of a barge platform and captured the flow separation induce by fluid viscous near the moonpool. They found that the moonpool significantly contributed to the viscous damping.

### 3. System-Level Studies

For the system-level studies of FOWTs, four aspects with increasing research complexity are presented: the simplified aerodynamics to examine the influence of aerodynamic loads on platform hydrodynamics, the prescribed platform motions to investigate the unsteady aerodynamic performance, the fully coupled aero-hydrodynamics and the aero-hydro-elastic behaviors considering the blade deformations.

### 3.1. Simplified Aerodynamics

The aerodynamic loads exerted on wind turbines are regarded as the quantities independent of time, with the aim of studying the platform hydrodynamics of FOWTs in a simplified scenario. Namely, the interference between the wind turbine and the floating platform in real time is not taken into account. Nematbakhsh et al. [123] studied the dynamic responses of a tension leg FOWT using a single-phase flow CFD method. The aerodynamic loads of wind turbine were simplified as the stationary thrust. Similarly, Zhao and Wan [124] investigated a OC4 semi-submersible platform under various wind speeds. They found that, for the low wind speed scenarios, the aerodynamic loads increase with wind speeds, and consequently, led to the significant dynamic responses of the platform. However, the simplified aerodynamics cannot reflect the strongly coupled feature between the wind turbine and the floating platform, so it is suited to be used in the preliminary analysis stage of hydrodynamic performance of FOWTs.

*3.2. Prescribed Platform Motions*

The aerodynamics of wind turbines affected by the platform motions is referred to as unsteady aerodynamic performance. The prescribed platform motions of surge, pitch and yaw are presented here due to their significant effects on the unsteady aerodynamics of wind turbines. The sine and cosine functions with amplitude and frequency are usually employed to define the prescribed platform motions. For example, the platform surge can be represented as:

$$X_{surge} = A cos\left(\frac{2\pi}{T}t\right) \tag{6}$$

where *A* is the surge amplitude, *T* is the surge period, *t* is the time. The velocity of surge motion can be expressed as:

$$U_{surge} = -A\left(\frac{2\pi}{T}\right)sin\left(\frac{2\pi}{T}t\right) \tag{7}$$

The change in relative wind speed due to platform surge will affect the aerodynamic performance of wind turbines, which can be expressed as:

$$U_{rel} = U_{ref} - U_{surge} = U_{ref} + \left(\frac{A2\pi}{T}\right)sin\left(\frac{2\pi}{T}t\right) \tag{8}$$

In order to analyze the effects of platform surge motion with various amplitudes and frequencies on the unsteady aerodynamic performance of wind turbines, Tran and Kim [125] performed the numerical simulations of a FOWT based on the unsteady BEM and CFD method with overset mesh technique. It was observed that the gap distances among the blade-up vortex tubes is variable due to contributions of platform surge motion and strong vortex-wake interactions, as shown in Figure 12. As the wind turbine moves backward or forward, these interaction effects tend to decrease or increase. In addition, the vortex-wake interactions are stronger with the increase in amplitude and frequency of surge motion. The results also showed that the pressure distributions on rotating blades are sensitive to the direction (forward or backward) of the platform surge. Chen et al. [126] utilized the IDDES turbulence model and the overset mesh technique in STAR CCM+ to investigate the effects of surge motion on the aerodynamics and wake instabilities of a FOWT. The flow separation on the blade suction surface was presented, and the separation position moving towards leading edge due to the larger relative wind speed induced by the higher frequency of surge motion. Kopperstad et al. [127] observed that a barge concept FOWT experiences faster wake recovery due to the large oscillation amplitude of the surge motion. Kyle et al. [128] investigated the vortex ring state of a FOWT under prescribed surge motion. The vortex ring state was observed due to the tip vortex interaction and the root vortex recirculation. Different from the above studies for a stand-alone FOWT with the prescribed surge motion, Rezaeiha and Micallef [129] studied the aerodynamic performance of two wind turbines based on the incorporation of the AD and the CFD method. The upstream one was oscillated with prescribed surge motion, and the downstream one was fixed and positioned 3D away from the upstream one. A low-frequency oscillating mode of the downstream rotor was observed, where the period approximately 10 times the surge period of upstream rotor.

For the prescribed pitch motion, Tran and Kim et al. [130] found that the aerodynamic power outputs of the wind turbine are 5.7 MW and 10 MW at the different pitch motion amplitudes of 1° and 4°, which are enhanced by 14% and 100%, respectively. However, Lei et al. [131] pointed out the effects of pitch motion on aerodynamic power of a vertical wind turbine are relatively smaller by employing the CFD method with IDDES turbulence model and overset mesh technique. Specifically, the results revealed that the increased amplitude and frequency of pitch motion are benefit to increase power coefficient, whereas the increase is less than 5%. Fang et al. [132] observed that amplitudes of rotor thrust and

torque decrease with the decreased frequency of pitch motion. Moreover, the blade-wake interactions are more evident with the higher amplitude and frequency of pitch motion.

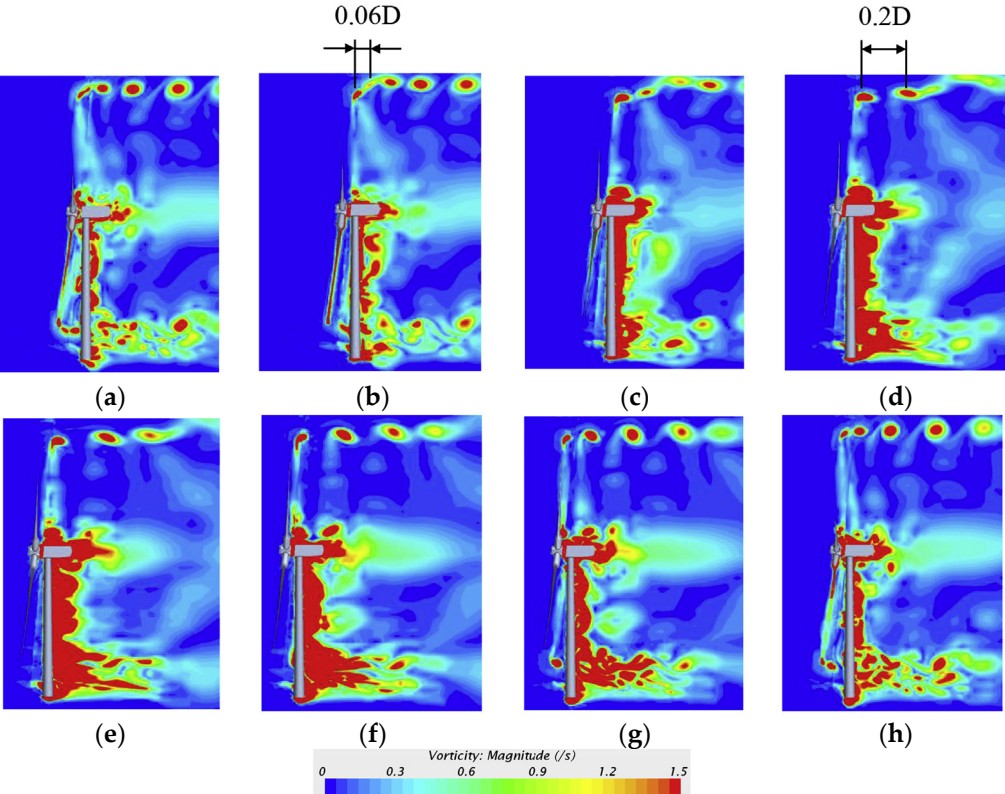

**Figure 12.** Visualization of instantaneous vorticity contours of the FOWT with the prescribed surge motion: (**a**) $t = 0T_{surge}$; (**b**) $t = 1/8T_{surge}$; (**c**) $t = 2/8T_{surge}$; (**d**) $t = 3/8T_{surge}$; (**e**) $t = 4/8T_{surge}$; (**f**) $t = 5/8T_{surge}$; (**g**) $t = 6/8T_{surge}$; (**h**) $t = 7/8T_{surge}$ [125]. Reproduced with permission from Tran and Kim, Renewable Energy; published by Elsevier, 2016.

Compared with the surge and pitch motions, the investigations of unsteady aerodynamic performance induced by platform yaw motion are relatively few. Cai et al. [133] performed the simulations of a FOWT with the prescribed yaw motion, the wind shear and tower shadow effects were considered. Leble and Barakos [134] pointed out that the discrepancies of rotor thrust and power between the fixed wind turbine and the floating one with yaw motion are 2.5%, indicating the insufficient influence of the yaw motion on unsteady aerodynamics. Liu et al. [135] studied the unsteady aerodynamics of a FOWT under three different platform motions including surge, pitch and heave, respectively.

The effects of surge and pitch motion on unsteady aerodynamics of wind turbine were compared by Li et al. [136] based on the unsteady AL in OpenFOAM. It was concluded that the effects of surge motion on aerodynamic performance are more obvious. Tran and Kim [88] outlined that the unsteadiness of wind turbine aerodynamics induced by platform pitch motion is approximately 12–16 times that of yaw motion. Lin et al. [137] performed the simulations of a FOWT with the coupled surge motion and pitch motion. The wake asymmetry was observed due to the complicated platform motions, and the wake expansion was promoted with the increase in wind speeds. Chen et al. [138] importantly noted that rotor power of the FOWT is reduced by 10% due to the coupled surge and pitch motions, compared to that of only the pitch motion.

Undoubtedly, in the initial design stage of FOWTs, it is suitable and valid to perform the FOWT behaviors with the prescribed platform motions. The underlying mechanisms of the wind turbine aerodynamics influenced by platform motions can be captured and analyzed through the unsteady characteristics of aerodynamic performance.

### 3.3. Fully Coupled Aero-Hydrodynamic Performance

Compared with the prescribed platform motions, the fully coupled aero-hydrodynamic behaviors of FOWTs excited by inflow wind and incident wave are more physical. In order to examine the contribution of aerodynamic loads to the dynamic responses of platform surge, Ren et al. [139] performed the simulations of a tension leg FOWT under the combined wind-wave conditions in Fluent. The surge motion was released, whereas the other 5-DOF were fixed. Liu et al. [140] investigated the dynamic responses of a semi-submersible FOWT with 3-DOF including surge, heave and pitch. The arbitrary mesh interface (AMI) and sliding mesh technique were employed to handle the relative movement between the rotating wind turbine and the platform. It was found that the mooring tension forces are significantly enhanced by the large response of the platform surge.

Regarding the 6-DOF simulations of FOWTs, Quallen and Xing et al. [97] employed a CFD solver CFDShip-Iowa V4.5 to address the 6-DOF motions of the FOWT. A variable-speed generator-torque controller is introduced to regulate the power generation. The increased mooring forces are beneficial to keep the FOWT in a more favorable variable-speed control region. Zhang and Kim [141] conducted a fully coupled CFD analysis for a semi-submersible FOWT using the STAR CCM+ incorporated with the overset mesh technique. Figure 13 shows the fluid domain of the FOWT, together with the mesh distribution. Their findings revealed that the rotor thrust is increased by 7.8% for the FOWT compared to the onshore wind turbine, whereas the rotor power is decreased by 10%, which may be attributed to the small windward area and the relative wind speed caused by tilted platform.

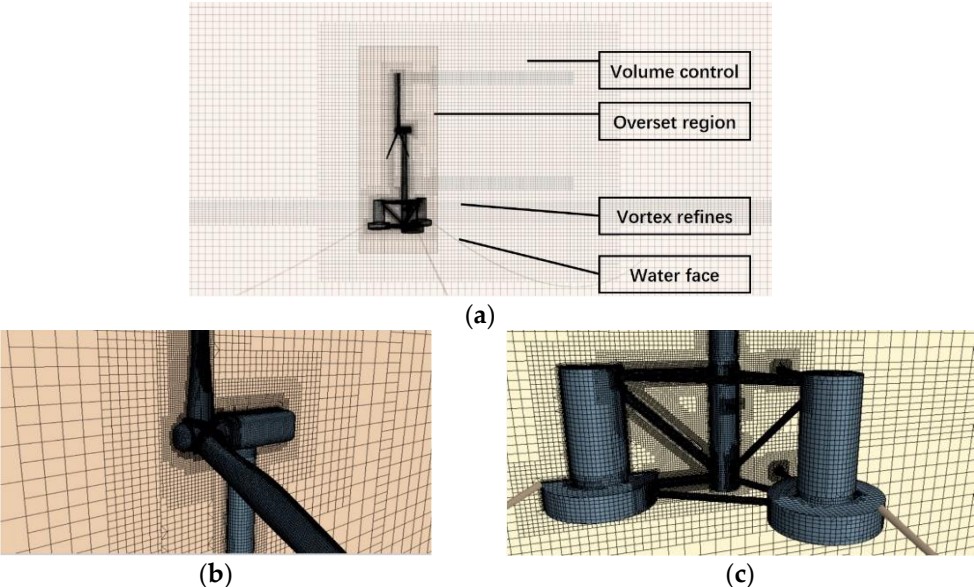

**Figure 13.** The computational domain of the FOWT: (**a**) mesh distribution of the FOWT; (**b**) mesh around wind turbine; (**c**) mesh around platform [141].

Tran and Kim [142] constructed a computational model to simulate the fully coupled aero-hydrodynamics of a semi-submersible FOWT. The results between CFD and the FAST code of surge response showed a large discrepancy of 19.6%, whereas the heave and pitch responses were closer among the two different frameworks. However, they particularly noted that unsteady aerodynamic performance of FOWTs predicted by the FAST code remains further investigations due to the great discrepancy when employing different aerodynamic methods. Specifically, the discrepancies of thrust coefficient for BEM and GDW are 24.0% and 33.3%, compared to that of the CFD. Additionally, the blade tip vortices and the vortices shedding from the hub, tower and platform are visualized, as shown in Figure 14. Zhou et al. [143,144] investigated the dynamic behaviors of a FOWT under three type incident waves, i.e., focused wave, irregular wave and reconstructed

focused wave. The hydrodynamic characteristics of the FOWT excited by irregular wave and reconstructed focused wave are similar, whereas they showed a great discrepancy compared to that of focused wave. In addition, tower bending moment and mooring tension forces presented dynamic responses at multiple frequencies, corresponding to the first, second and higher-order frequencies of the natural frequency of the structure, indicating the nonlinear properties of the system.

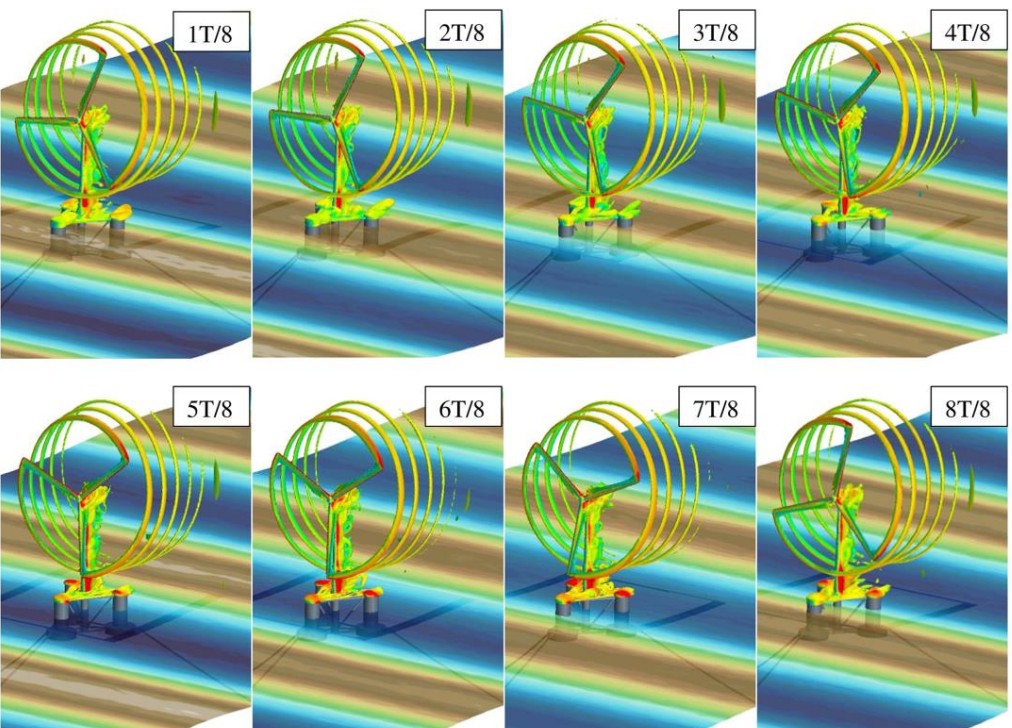

**Figure 14.** Instantaneous iso-vorticity contours for the semi-submersible FOWT with 6-DOF motions [142].

However, the blade-resolved modeling for the numerical simulations of fully coupled aero-hydrodynamics of FOWTs are computationally expensive. In order to save the computational costs, Cheng et al. [106] developed a novel CFD solver FOWT-UALM-SJTU by incorporating the unsteady AL considering the platform motions and an in-house CFD solver naoe-FOAM-SJTU for floating platform hydrodynamics. Figure 15 shows the implement of the unsteady AL to the in-house CFD solver naoe-FOAM-SJTU. The FOWT composed of NREL 5 MW wind turbine and a semi-submersible platform was employed as the research object, and its coupled responses were investigated.

Huang et al. [145,146] also employed the unsteady AL to represent the wind turbine based on the developed CFD solver FOWT-UALM-SJTU. The interference effects between wind turbine and spar buoy platform under combined wind-wave excitation were fully investigated by performing cases with various platform DOF and turbine states. As shown in Figure 16, the blade tip vortex is captured and visualized. Moreover, the platform surge and pitch responses increase significantly due to the aerodynamic loads exerted on the wind turbine, whereas the heave response is reduced due to the vertical component of rotor thrust. Zheng et al. [104] also employed the unsteady AL to conducted the coupled dynamic responses of a spar buoy FOWT with heave plate.

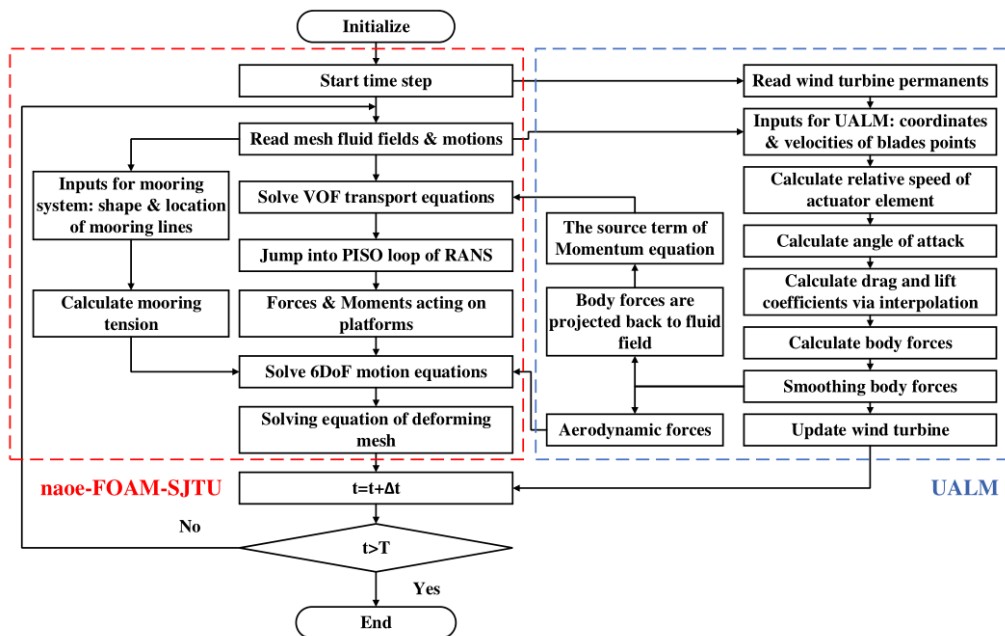

**Figure 15.** The implement of the unsteady AL to the in-house CFD solver naoe-FOAM-SJTU [145].

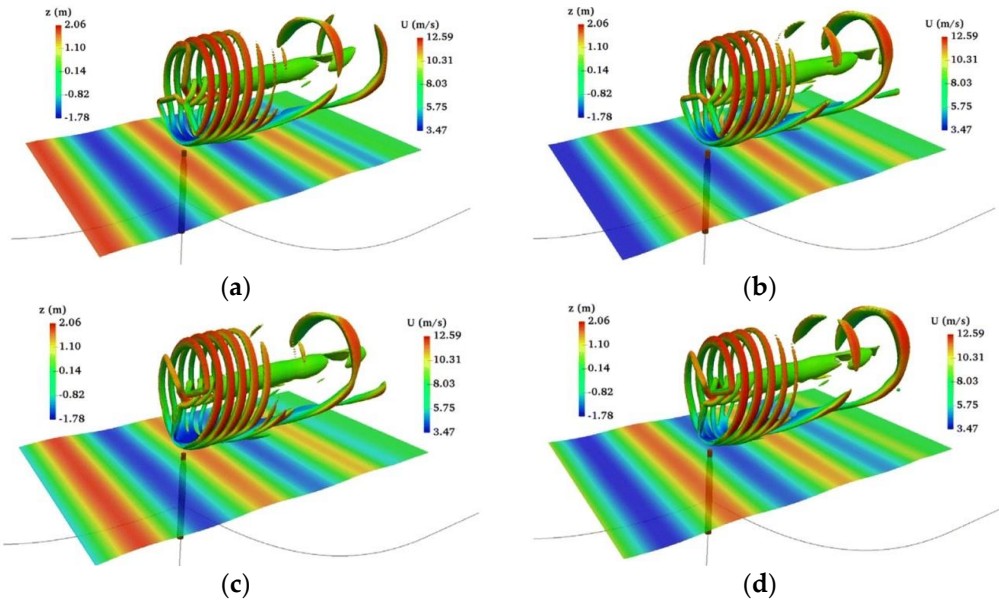

**Figure 16.** Instantaneous vortex structures of the spar buoy FOWT modeled by the unsteady AL: (**a**) $t = 0$; (**b**) $t = 1/4T_{wave}$; (**c**) $t = 2/4T_{wave}$; (**d**) $t = 3/4T_{wave}$ [146]. Reproduced with permission from Huang et al., Journal of Marine Science and Application; published by Springer Nature, 2019.

### 3.4. Fully Coupled Aero-Hydro-Elastic Performance

In order to investigate the rotor-wake interactions of FOWTs, Steven et al. [147] integrated the FVM to an aeroelastic numerical framework with the strongly two-way coupled communications between fluid and structure domains. Due to the lack of aeroelastic experimental measurements for FOWTs, the validations were conducted separately against the aerodynamic performance and structure deformations of experimental data. After that, the validated aeroelastic framework was applied to study the rotor-wake interactions and the related aeroelastic characteristics of FOWTs under a series of operational conditions [148]. Although the FVM is capable of accurately predicting the aerodynamics of wind turbines under the complex operational conditions, the flow separation on the blade surfaces is not reasonably captured due to the absence of fluid viscosity. Its applications

for FOWT aerodynamics need further validation, particularly for the situations with more complex aeroelastic simulations. Hence, a high-fidelity CFD analysis tool is necessary for the investigations of aero-hydro-elastic performance of FOWTs.

By combining the high-fidelity CFD solver developed for FOWT aero-hydrodynamics and a structure solver MBDyn, Liu et al. [149] investigated the aeroelastic characteristics of a FOWT with prescribed surge motion. Specifically, the blade structure deformations are solved by the structure solver MBDyn based on the forces exert on turbine blades which is predicted by the fluid solver pimpleDyMFoam. Subsequently, the blade deformations are delivered to the fluid solver and represented through the updated mesh. In addition to demonstrate the negative effects of blade deformations on aerodynamic performance, they also conducted a simulation to study how the platform surge motion affects the dynamics of flexible turbine blades. It was found that the variations of rotor thrust and power are enhanced when imposing the surge platform motion. After that, they developed a fully coupled aero-hydro-elastic tool for the FOWTs under combined wind-wave conditions [150]. The capability of the developed tool was demonstrated by a semi-submersible FOWT dynamic responses in terms of blade deformations, aerodynamic performance, platform motions and tension forces of mooring cables.

Due to the computationally expensive costs for the blade-resolved CFD simulations, Huang et al. [151,152] developed an aero-hydro-elastic framework for FOWTs by integrating the AL to in-house CFD code naoe-FOAM-SJTU. The AL was modified to account for the induced velocities caused by platform motions and blade deformations. The blade structures were represented by the beam model and discretized by the 1D FEM. The proposed framework was validated by previous published numerical results. Then they investigated the dynamic responses of a stand-alone FOWT and the two tandem-arranged FOWTs with blade deformations. The results showed that the time-averaged values of rotor power and thrust are decreased due to the presence of blade deformations, whereas the fluctuation amplitudes are enhanced. The blade deformations of downstream FOWT are smaller than that of upstream FOWT, and more stable vortexes are found when the blade deformations are presented.

## 4. Future Recommendations

### 4.1. Fully Coupled Aero-Hydro-Elastic Performance

The FOWT is a complex multi-integrated system consisting of wind turbine blades, hub, nacelle, tower, floating platform and mooring system. The design and development of FOWTs requires multi-disciplinary knowledge, and the strong interactions between components are significant. As mentioned in Section 3.3, a lot of efforts are dedicated to investigate the fully coupled aero-hydrodynamics of FOWTs, which is a sophisticated and challengeable issue for the high-fidelity CFD simulations and essential for the aero-hydro-elastic behaviors. However, to the best of the author's knowledge, the CFD research for the aero-hydro-elastic performance of FOWTs is so scarce that the aeroelastic characteristics of FOWTs are not systematically examined and analyzed. Therefore, some efforts should shift to the aero-hydro-elastic performance of FOWTs, particularly for the situations of large-scale blades and severe sea conditions.

The structure models can be introduced into the aero-hydrodynamic framework to account for the blade deformations of FOWTs. The 3D FEM incorporated with the blade-resolved modeling for aerodynamics is the highest-fidelity aeroelastic analysis method, with it, the detailed stress and strain of blade structures are captured. However, it is noteworthy that mismatch between fluid domain mesh and structural domain mesh, mesh update due to blade deformations and computational costs are the matters of focus. The 1D EBM incorporated with the blade-resolved modeling for FOWTs is an option to save the computational resources to some extent. Furthermore, the blade-resolved modeling replaced by the AL and combined with the 1D EBM namely the EAL aeroelastic framework is the most suitable choice for saving the computational costs in the CFD-related simulations of aero-hydro-elastic behaviors of FOWTs. Overall, the desired results and the affordable

computational costs are the main considerations when selecting the aerodynamic model and structure model for the construction of aero-hydro-elastic framework of FOWTs.

### 4.2. Complex Atmospheric Inflow

The development of FOWTs towards large-scale is unstoppable, leading to the more complex atmospheric inflow conditions over the wind turbine rotor. One of the distinct features for atmospheric inflow is the wind shear, which is characterized by the increasing wind speed with height. Although the wind shear can be taken into account in some studies of high-fidelity CFD simulations for FOWTs [145,153], it is still an oversimplified situation. Specifically, the turbulence characteristics of atmospheric inflow, highly correlated with the fatigue loads and structure failure of wind turbines, are not considered. Moreover, due to the absent contribution of large-scale atmospheric turbulence, the wake meandering effects are not reproduced, which are characterized by significant lateral oscillations and have a remarkable impact on inflow conditions of downstream wind turbines.

Two turbulence models are suggested by International Electrotechnical Commission (IEC) standard to generate the atmospheric inflow conditions for wind turbines, the Mann spectral tensor model [154], hereafter denoted "Mann model", and the Kaimal spectral and exponential coherence model [155], hereafter denoted "Kaimal model". Li et al. [156] investigated the effects of atmospheric inflow conditions on the aerodynamics of a semi-submersible FOWT based on the Kaimal model, the results revealed that the rotor power is more unstable due to the presence of atmospheric turbulence. However, the above two engineering models are concluded from small onshore wind turbines, their applications for large FOWTs needs further validation. By generating the atmospheric inflow conditions for spar buoy FOWT based on the two engineering models and high-fidelity LES, Doubrawa et al. [157] noted that fatigue loads of the spar buoy FOWT in high-wind scenarios are overpredicted by the two engineering models and vice versa. Nybø et al. [158] also emphasized that the aforementioned two engineering models may lead to incorrect estimations for FOWT dynamic responses. Moreover, the atmospheric stability is not taken into account in the two engineering models, which is recognized as a key factor for the wake recovery of wind turbines. Consequently, some efforts should be focused on the LES to generate more reasonable and realistic atmospheric inflow conditions and to further validate the applicability of the above two engineering models for FOWTs.

### 4.3. Wake Interactions between Multi-FOWT

In reality, the FOWTs are presented in the form of a floating wind farm with the purpose of commercial operation. Due to the constraints of sea area and mooring cables, the downstream FOWTs will inevitably operate in the wakes of upstream FOWTs, called the wake effects, which may reduce the power outputs and increase the fatigue loads. Additionally, the floating wind farm suffers more power deficit attributed by low turbulence intensity of high-quality wind resources, compared to that of an onshore wind farm. Therefore, the investigations for muti-FOWT are required to carry out the underlying physical mechanisms of wake interactions, for the purpose of reducing the power deficit and fatigue loads of floating wind farm.

Rezaeiha and Micallef [129] studied the wake interactions of two tandem FOWTs using the CFD analysis incorporated with the AD. Three different surge amplitudes of upstream FOWT were carried out and the power performance of the two floating rotors as well as the wake interactions were examined. It was found that the mean power outputs of the both floating rotors are slightly enhanced, and a faster wake recovery is observed due to the enhanced flow mixing caused by surge motion of upstream FOWT. Recently, Zhang et al. [159] conducted a comprehensive study for two FOWTs based on the blade-resolved modeling, the rotor power, torque and platform motions were analyzed. However, the studies for wake interactions between multi-FOWT are still particularly rare. For the deployment of commercial floating wind farm, more detailed investigations for the wake effects between multi-FOWT are urgently required.

## 5. Conclusions

In this study, high-fidelity CFD simulations for the dynamic responses of FOWTs are comprehensively summarized and analyzed. The component-level studies including aerodynamics, aeroelasticity and hydrodynamics are presented. The system-level studies consisting of simplified aerodynamics, prescribed platform motions and fully coupled aero-hydrodynamics are performed, as well as a rare studies for aero-hydro-elastic performance of FOWTs. In addition to the fully coupled aero-hydro-elastic behaviors of FOWTs, another two future research directions are suggested, i.e., the complex atmospheric inflow and the wake interactions of multi-FOWT.

For the wind turbine aerodynamics, the blade-resolved modeling is the most precise method, where the detailed flow field on blade surfaces is captured, but suffers expensive computational costs. The parametric modeling of wind turbine blades is considered to significantly save computational resources, which is referred to actuator models and composed of the AD, the AL and the AS. Among the three actuator models, the AL is incorporated with the 1D EBM for the aeroelastic analysis of wind turbines. The AL-related aeroelastic framework can reasonably predict the blade deformations under the specific and normal cases. However, if more detailed results of blades deformations are desired, such as the stress and strain of blade structures, the blade-resolved modeling with the 3D FEM is suggested. The hydrodynamic studies of floating platforms are more focused on loads and motion responses induced by incident waves and vortices, and the nonlinear characteristics of dynamic responses are captured.

The research of simplified aerodynamics used to study the effects of aerodynamic loads on dynamic responses of floating platforms is comparatively few, compared to that of prescribed platform motions to investigate the unsteady performance of wind turbine aerodynamics. Among the 6-DOF of floating platforms, the prescribed surge and pitch motions with various amplitudes and frequencies are extensively studied, as well as the yaw motion occasionally. Subsequently, for a more physical situation, the combined wind-wave conditions are employed for the fully coupled aero-hydrodynamic performance of FOWTs. The structure model is introduced into the aero-hydrodynamic framework with the aim of considering the blade deformations of FOWTs. However, the relevant research for high-fidelity CFD simulations of the aero-hydro-elastic behaviors of FOWTs is still rare.

In order to capture more wind energy and reduce the LCOE, the scale of FOWTs becomes larger. However, some issues are significantly caused by the large rotor diameters. One is the blade structure deformations which is not systematically examined and analyzed in the high-fidelity CFD frameworks for FOWTs. Another is the modeling of more realistic atmospheric inflow by LES rather than the employment of engineering synthetic wind model. Moreover, the complex wake interactions between multi-FOWT should be investigated to figure out the underlying mechanisms of wake effects and to improve the overall power outputs of the multi-FOWT. Consequently, in order to improve the operational performance and structural reliability of FOWTs, as well as the commercialization of floating wind farms, the high-fidelity investigations for aero-hydro-elastic behaviors of FOWTs, modeling of more realistic atmospheric and wake interactions of multi-FOWT are indispensable.

**Author Contributions:** Conceptualization, S.X., Y.X., W.Z., D.W.; writing—original draft preparation, S.X. and Y.X.; writing—review and editing, W.Z., D.W.; supervision, D.W. All authors have read and agreed to the published version of the manuscript.

**Funding:** The work is supported by the National Key Research and Development Program of China (Grant No. 2019YFB1704200), and National Natural Science Foundation of China (Grant Nos. 51909160, 52131102 and 51879159).

**Institutional Review Board Statement:** Not applicable.

**Informed Consent Statement:** Not applicable.

**Data Availability Statement:** Not applicable.

**Conflicts of Interest:** The authors declare no conflict of interest.

## Nomenclature

| | |
|---|---|
| GWEC | Global Wind Energy Council |
| FOWTs | floating offshore wind turbines |
| LCOE | levelized cost of energy |
| HPC | high-performance computers |
| BEM | blade element momentum |
| PF | potential flow |
| CFD | computational fluid dynamics |
| AD | actuator disk |
| AL | actuator line |
| AS | actuator surface |
| 3D | three-dimensional |
| LES | Large Eddy Simulation |
| ACE | actuator curve embedding |
| RANS | Reynolds-Averaged Navier-Stokes |
| uRANS | unsteady Reynolds-Averaged Navier-Stokes |
| DES | Detached Eddy Simulation |
| IDDES | Improved and Delayed Detached Eddy Simulation |
| DDES | Delayed Detached Eddy Simulation |
| FVM | free vortex wake |
| FEM | finite element method |
| EBM | equilibrium beam model |
| MBD | multi-body dynamics |
| GEBT | geometrically exact beam theory |
| CSD | computational structure dynamics |
| EAL | elastic actuator line |
| FSI | fluid-structure interaction |
| ME | Morison equation |
| 6-DOF | six degrees-of-freedom |
| FDM | finite difference model |
| FSM | finite segment model |
| FEM | finite element model |
| LMM | lumped mass model |
| MBS | multi-body system |
| VIM | vortex-induced motions |
| AMI | arbitrary mesh interface |
| IEC | International Electrotechnical Commission |

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
