# Peer review of "A Review of High-Fidelity Computational Fluid Dynamics for Floating Offshore Wind Turbines"

_jmse, doi:10.3390/jmse10101357_

Round 1

Reviewer 1 Report

In this nicely written manuscript, the authors present an overarching review of high-fidelity CFD methods for modelling floating offshore wind turbines. The proposed work fits within the aim of the journal and presents a topic of high interest. Furthermore, I would like to congratulate the authors for the excellent quality of the review article. However, there are some errors that should be addressed before its publication. Hence, I recommend the publication of the manuscript once the following points have been addressed:

1.    Acronyms should be avoided in the abstract.

2.    The introduction section should be improved, and the novelty of this review article justified. This should be done in the frame of other previous review articles.

3.    A list of acronyms should be included in the manuscript.

4.    I miss the mention of any Lagrangian method, such as SPH (Smoothed Particle Hydrodynamics).

The quality of the English and the fluency of the text is, in general, good, however, the manuscript t would benefit from a review by a native speaker, as there are some sentences with some grammar errors (e.g., line 94).

Reviewer 2 Report

Dear Authors,

 The manuscript is a good collection of research work done on FOWTs using CFD and multiphysics interaction. It details the mechanisms, results and flow phenomena with various examples at several fidelity.

There are few suggestions from my side which will improve the content and complete the review.

1) Could you add a section on underwater noise generation for FOWT solo and farm? Spectral analysis of noise signature will show how many decibels each unit and a farm generates. This will be crucial to know where to place them.

2) Cross wind operation is also an important metric for these farms. A paragraph about these efforts must be included with few details on design improvements, optimizations to take advantage of cross winds.

3) A paragraph on FOWTs in extreme weather conditions must be included since it is very often these farms will experience them and yaw errors have a huge effect on mooring mechanisms.
